# L-Proline Prevents Endoplasmic Reticulum Stress in Microglial Cells Exposed to L-azetidine-2-carboxylic Acid

**DOI:** 10.3390/molecules28124808

**Published:** 2023-06-16

**Authors:** Jordan Allan Piper, Nour Al Hammouri, Margo Iris Jansen, Kenneth J. Rodgers, Giuseppe Musumeci, Amolika Dhungana, Sahar Masoumeh Ghorbanpour, Laura A. Bradfield, Alessandro Castorina

**Affiliations:** 1Laboratory of Cellular and Molecular Neuroscience (LCMN), School of Life Sciences, Faculty of Science, University of Technology Sydney, P.O. Box 123, Broadway, Sydney, NSW 2007, Australia; jordan.piper@uts.edu.au (J.A.P.); nour.alhammouri@student.uts.edu.au (N.A.H.); margo.jansen@student.uts.edu.au (M.I.J.); 2Neurotoxin Research Group, School of Life Sciences, Faculty of Science, University of Technology Sydney, P.O. Box 123, Broadway, Sydney, NSW 2007, Australia; kenneth.rodgers@uts.edu.au; 3Department of Biomedical and Biotechnological Sciences, Section of Anatomy, Histology and Movement Science, School of Medicine, University of Catania, Via S. Sofia n°97, 95123 Catania, Italy; g.musumeci@unict.it; 4School of Life Sciences, Faculty of Science, University of Technology Sydney, P.O. Box 123, Broadway, Sydney, NSW 2007, Australia; amolika.dhungana@uts.edu.au (A.D.); masoumeh.s.ghorbanpour@student.uts.edu.au (S.M.G.); laura.bradfield@uts.edu.au (L.A.B.)

**Keywords:** azetidine-2-carboxylic acid, unfolded protein response, endoplasmic reticulum, microglia, multiple sclerosis, L-proline, neuroinflammation

## Abstract

L-Azetidine-2-carboxylic acid (AZE) is a non-protein amino acid that shares structural similarities with its proteogenic L-proline amino acid counterpart. For this reason, AZE can be misincorporated in place of L-proline, contributing to AZE toxicity. In previous work, we have shown that AZE induces both polarization and apoptosis in BV2 microglial cells. However, it is still unknown if these detrimental effects involve endoplasmic reticulum (ER) stress and whether L-proline co-administration prevents AZE-induced damage to microglia. Here, we investigated the gene expression of ER stress markers in BV2 microglial cells treated with AZE alone (1000 µM), or co-treated with L-proline (50 µM), for 6 or 24 h. AZE reduced cell viability, nitric oxide (NO) secretion and caused a robust activation of the unfolded protein response (UPR) genes (ATF4, ATF6, ERN1, PERK, XBP1, DDIT3, GADD34). These results were confirmed by immunofluorescence in BV2 and primary microglial cultures. AZE also altered the expression of microglial M1 phenotypic markers (increased IL-6, decreased CD206 and TREM2 expression). These effects were almost completely prevented upon L-proline co-administration. Finally, triple/quadrupole mass spectrometry demonstrated a robust increase in AZE-bound proteins after AZE treatment, which was reduced by 84% upon L-proline co-supplementation. This study identified ER stress as a pathogenic mechanism for AZE-induced microglial activation and death, which is reversed by co-administration of L-proline.

## 1. Introduction

In addition to the 20 canonical amino acids found in nature, there are several non-protein amino acids (NPAAs) that do not contribute to protein synthesis. These NPAAs are highly abundant in plants and are well known for their role in deterring predators via a biological phenomenon called allelopathy, as well as acting as reservoirs for nitrogen storage [1,2]. L-Azetidine-2-carboxylic acid (AZE) is one such NPAA, and it has been detected in human food sources, especially in sugar and table beets [3,4,5].

AZE was firstly identified by Rubenstein in 2006 [5]. Upon its identification, Rubenstein developed a novel unifying hypothesis linking AZE consumption to multiple sclerosis (MS) pathogenesis, in which geographical and historical aspects of the disease were also taken into account [6]. Later studies revealed that the identified NPAA triggers specific pathological alterations in at least two resident cell populations of the central nervous system (CNS), oligodendrocytes and microglia [7,8], corroborating the initial pathogenic link with MS. AZE exhibits high structural similarity with L-proline [9]. For this reason, upon its entry into cells, AZE can evade recognition by aminoacyl-tRNA synthetases (aaRS) and be misincorporated into the sequence of proline-rich proteins upon consumption [6,10]. AaRS enzymes normally attach each amino acid to its cognate tRNA, which are then transferred to the ribosome and utilized as building blocks for protein synthesis. These synthetases evolved an editing activity that either removes misactivated amino acids or hydrolytically clears the misattached amino acids from the mischarged tRNA, thereby controlling the accuracy of protein synthesis [11]. AZE fits into the active site pocket of prolyl-tRNA synthetase (ProRS) and is activated for protein synthesis, thus it can be erroneously incorporated into L-proline positions during protein assembly [9]. The resulting misincorporation is believed to increase the immunogenicity of affected proteins and promote protein misfolding and endoplasmic reticulum (ER) stress in intoxicated cells [12], potentially contributing to the development of autoimmune or other neurological disorders [6].

L-Proline residues constitute nearly 6% of the human proteome [13]. Proline-rich proteins are essential for collagen production and for the synthesis of salivary proteins [14]; however, they are also abundant in the CNS [15], especially in core myelin proteins and other myelin constituents that are crucial for axonal cytoskeletal stabilisation and the control of axon-myelin interactions, such as the myelin associated glycoprotein [16,17]. Therefore, it is conceivable that AZE misincorporation may contribute to the pathogenesis of both myelin and perhaps non-myelin associated CNS diseases. Sobel’s recent study supports this claim, as the authors demonstrated the dose-dependent detrimental effects of AZE administration on rodents’ myelin health, since the toxic NPAA caused severe oligodendrogliopathy and microglial clustering in the CNS white matter [8]. In line with these findings, our laboratory has recently explored the effects AZE exposure in the murine BV2 microglial cell line, where we revealed novel and potent pro-apoptotic and pro-inflammatory effects of the NPAA [7]. These effects are particularly relevant in the context of neuroinflammatory diseases, including MS, where chronic microglial polarization is known to contribute to both neuronal damage [18,19,20] and demyelination [21]. Based on these findings, it is reasonable to assume that AZE-induced CNS pathology could also be partaken by the cumulative misfolding events affecting the several proline-rich proteins within the CNS, thereby disrupting the biological activities of microglia (and possibly other CNS cell types), thus culminating in neuroinflammation and neurodegeneration. Unfortunately, the specific mechanisms responsible for microglial pathology in AZE-treated cells have not been explored. As recently proposed by Roest et al. (2018) [12] and Sobel et al. (2022) [8], one pathogenic mechanism could be that AZE substitution of L-proline during protein assembly induces protein misfolding in several other glial cells (in addition to oligodendrocytes), resulting in diffuse ER stress in the CNS. Interestingly, eukaryotic cells are embedded with an evolutionary-conserved protective mechanism designed to prevent the excessive accumulation of misfolded proteins and restore ER homeostasis, called the unfolded protein response (UPR) [22]. The UPR also coordinates the inflammatory responses of resident microglia [23,24]. This emphasizes the potential role of the UPR in regulating microglial activation upon AZE exposure—which may contribute to increase the neuroinflammatory burden. To test this theory, in the present study we used an in vitro model of AZE toxicity to define the molecular mechanisms underlying AZE-mediated cell polarisation. Additionally, given the structural similarities existing between AZE and L-proline, we postulated that L-proline could potentially outcompete AZE during protein assembly, thereby leading to reduced accumulation of misfolded proteins and subsequently mitigating ER stress. To address this further hypothesis, microglial cells were co-administered with AZE and L-proline, and then tested through a series of biological and molecular assays to establish whether this intervention could alleviate the detrimental effects of the NPAA in our in vitro toxicity model. The primary goal of this study is to provide proof-of-concept data to highlight how a plant-sourced NPAA can trigger ER stress and inflammation in microglial cells, and how simple dietary interventions (L-proline supplementation) may be critical in preventing and perhaps reversing the noxious effects of environmental exposure to this toxicant and its putative damaging effects on CNS health.

## 2. Results

### 2.1. L-Proline Supplementation Prevents the Detrimental Effects of AZE Intoxication on Cell Viability and Nitric Oxide (NO) Release in BV2 Microglial Cells

MTT assay was used to assess whether L-proline co-supplementation elicited any beneficial effects on cell viability and NO release in cells undergoing the same toxicity regime used in our previously established model of AZE toxicity [7]. As depicted in Figure 1A, AZE (1000 µM) caused a progressive and significant reduction in BV2 microglia viability (** *p* < 0.01 at 6 h and **** *p* < 0.0001 at 24 h vs. control, respectively). Co-administration of L-proline at a concentration 20× lower than AZE (50 µM) was sufficient to fully prevent cell viability at both experimental time points (# *p* < 0.05 and #### *p* < 0.0001 vs. AZE treated cells; Figure 1A).

In agreement with previous findings, NO release—measured using the Griess assay and used as a measure of the inflammatory response of microglia—demonstrated that AZE caused a robust inflammatory response both at 6 and 24 h post-treatment (**** *p* < 0.0001 vs. control at both time points; Figure 1B). L-proline supplementation fully abrogated AZE-driven NO release at the experimental time points tested (#### *p* < 0.0001 vs. AZE-treated cells; Figure 1B), with NO reaching levels comparable to that seen in untreated controls.

### 2.2. L-Proline Reverses the Expression of Pro- and Anti-Inflammatory Genes in AZE-Intoxicated BV2 Microglia

To elucidate the mechanisms underlying AZE-induced inflammatory profile, and assess whether these could be reversed by L-proline supplementation, we examined the expression of common pro- (IL6 and IBA1) and anti-inflammatory genes (CD206) in BV2 microglia exposed to AZE only or in combination with L-proline after 1, 6 and 24 h by real-time quantitative polymerase chain reaction (qPCR). Additionally, we also investigated the expression levels of TREM2 mRNAs, whose expression is repressed in microglia upon exposure to pro-inflammatory stimuli [25]. As illustrated in Figure 2A, AZE treatment reliably increased the gene expression of the pro-inflammatory cytokine IL6 already after 1 h treatment (*** *p* < 0.001 vs. control; Figure 2A), which further increased (> 600 folds) at 6 and 24 h (*** *p* < 0.001 and **** *p* < 0.0001 vs. control, respectively). Whilst L-proline co-administration was unable to prevent AZE-driven IL6 induction at 1 h (*p* > 0.05), it remarkably diminished IL6 mRNAs at 6 h (*** *p* < 0.001 vs. AZE) and at 24 h (**** *p* < 0.0001 vs. AZE; Figure 2B). IBA1 gene expression was not increased in response to AZE after 1 h (*p* > 0.05); however, gene expression was significantly up-regulated at 6 h (** *p* < 0.01 vs. control; Figure 2C), but not at 24 h (*p* > 0.05 vs. control) corroborating our previous data [7]. Accordingly, L-proline completely prevented IBA1 up-regulation caused by AZE treatment at 6 h (* *p* < 0.05 vs. AZE; Figure 2D), with minor and non-significant effects at 1 and 24 h (*p* > 0.05 vs. AZE). We also found that TREM2 expression was reduced by AZE at both 1 and 24 h (* *p* < 0.05 vs. control for both; Figure 2E), but not at 6 h (*p* > 0.05). L-proline addition to the culture media normalized the expression of TREM2 to control levels both at 1 and 6 h (*p* > 0.05 vs. AZE), and significantly increased TREM2 transcripts at 24 h (* *p* < 0.05 vs. AZE; Figure 2F).

Measurements of CD206, an anti-inflammatory marker, revealed a slight reduction in transcript levels after 1 h (*p* = 0.07 vs. control), which was statistically significant at 6 and 24 h (* *p* < 0.05 vs. control for both; Figure 2G). Surprisingly, co-treatment with L-proline had little to no effect on the expression of CD206 (*p* > 0.05 vs. AZE at all time points; Figure 2H). Comparative analyses demonstrating lack of effects of L-proline alone on the expression of pro- and anti-inflammatory genes in cells are provided in Appendix A.

### 2.3. AZE Treatment Causes ER Stress and Activation of UPR Genes

To establish if AZE intoxication promoted the induction of ER stress genes, we conducted time-course analyses of a panel of genes known to be involved in the activation of ER stress pathways [26].

Expression of the upstream activator of the ER stress response ATF6 was significantly downregulated after 1 h of treatment with AZE (* *p* < 0.05 vs. control; Figure 3A); however, expression levels were rapidly increased after 6 h (** *p* < 0.01) to then normalize to control levels after 24 h (*p* > 0.05). A similar behavior was observed when measuring PERK gene (also known as EIF2AK3) expression, where we observed a reduction in mRNA levels after 1 h (* *p* < 0.05 vs. control; Figure 3B), followed by an increase at 6 h (** *p* < 0.01) and return to control levels at 24 h (*p* > 0.05). Consistent with both ATF6 and PERK transcriptional responses to AZE insults, IRE1α (gene name: ERN1) also showed an early down-regulation at 1 h (* *p* < 0.05 vs. control; Figure 3C), followed by a transient up-regulation at 6 h (* *p* < 0.05) to then return to baseline (control) levels within 24 h (*p* > 0.05).

In line with the above results, key downstream effectors of the ER stress/UPR pathways (ATF4, GADD34, XBP1 and DDIT3; please refer to the diagram in Figure 3) were globally up-regulated after AZE treatment (Figure 3D–G). Specifically, ATF4 mRNAs were significantly increased starting from 6 h (** *p* < 0.01 and * *p* < 0.05 vs. control at 6 and 24 h, respectively; Figure 3D). GADD34 gene expression was not affected until the 24 h time point, where we report a significant increase (* *p* < 0.05 vs. control; Figure 3E). The XBP1 gene was increased as early as after 1 h post-AZE treatment (* *p* < 0.05 vs. control; Figure 3F) and was followed by a progressive increase at 6 h (** *p* < 0.01) and 24 h (*** *p* < 0.001). Finally, DDIT3 gene expression was increased by AZE treatment after 1, 6 and 24 h (* *p* < 0.05 at all of the time points tested; Figure 3G).

### 2.4. L-Proline Treatment Prevents ER Stress and the Induction of UPR Genes

Once we determined that AZE caused ER stress and consequently triggered the activation of UPR genes, we aimed to assess if L-proline co-treatment prevented these effects.

Whilst L-proline caused a further but marginal increase in ATF6 gene expression in AZE-treated cells after 1 h (*p* > 0.05 vs. AZE; Figure 4A), it remarkably reduced AZE-induced gene expression after 6 h (** *p* < 0.01), but not after 24 h (*p* > 0.05). The up-regulation of PERK gene expression caused by AZE was also significantly diminished in the presence of L-proline after 6 (** *p* < 0.01) and 24 h (*** *p* < 0.001) (Figure 4B). A similar pattern of gene expression was seen when interrogating IRE1α, with a non-significant up-regulation after 1 h (*p* > 0.05 vs. AZE; Figure 4C), followed by a significant down-regulation of IRE1α after 6 and 24 h (* *p* < 0.05; Figure 4C).

Accordingly, downstream UPR effector genes were also reduced upon L-proline co-administration (Figure 4D–G). Specifically, after inducing an early ATF4 gene up-regulation at 1 h (* *p* < 0.05 vs. AZE; Figure 4D), L-proline co-supplementation strongly reduced ATF4 mRNAs at both 6 and 24 h (**** *p* < 0.0001) (Figure 4D). In contrast, L-proline had no effects on AZE-driven GADD34 mRNAs at 1 and 6 h (*p* > 0.05), whose expression levels were only reduced at the latest time point tested (* *p* < 0.05 vs. AZE; Figure 4E). XBP1 gene expression also showed a similar pattern of gene regulation as GADD34, with a significant reduction recorded only after 24 h L-proline co-treatment (*** *p* < 0.001 vs. AZE; Figure 4F). DDIT3 gene expression was unaffected by L-proline co-administration at 1 h (*p* > 0.05 vs. AZE; Figure 4G) but was robustly reduced after 6 h (*** *p* < 0.01) and even more so after 24 h (**** *p* < 0.0001; Figure 4G).

Our analyses also revealed some intrinsic effects of L-proline treatment alone on the expression of downstream regulators of the ER stress/UPR cascade in the BV2 microglial cell line, the results of which are shown in Appendix A.

### 2.5. L-Proline Prevents AZE-Driven Induction of ER Stress Activator PERK in BV2 Microglial Cells

To complement our transcriptional findings, we also conducted immunofluorescence (IF) studies in BV2 microglia following treatment with AZE alone or in combination with L-proline for 24 h. Untreated cells (control) and L-proline alone (drug-treatment control) were also included as internal controls.

As depicted in the representative photomicrographs shown in Figure 5A, and as confirmed by IF quantification, AZE treatment significantly increased PERK immunoreactivity (** *p* < 0.01 vs. control; Figure 5A,B). The co-presence of L-proline in the culture media was sufficient to prevent AZE-driven PERK induction, as demonstrated by a significant reduction in IF intensity in AZE + L-proline treated cells (# *p* < 0.05 vs. AZE; Figure 5A,B). No significant effects were seen in cells treated with L-proline alone.

### 2.6. L-Proline Co-Treatment Prevents IBA1 Induction and Phosphorylation of the ER Stress Transducer Phospho-IRE1^(Ser724)^ in Primary Microglia Exposed to AZE

To confirm our findings, indicating that AZE can induce both inflammation and activate ER stress response mechanisms, and validate our theory that L-proline is able to prevent such detrimental effects, we carried out co-IF studies in murine primary microglia using two primary antibodies raised against IBA1 and phospho-IRE1α^(Ser724)^. We specifically stained cells with phospho-IRE1α, as phosphorylation of the ER stress transducer IRE1α is required to initiate the transmittance of the unfolded protein signal from the ER to the nucleus [27].

As shown in Figure 6, AZE treatment caused a robust induction of IBA1-immunoreactivity in primary microglia (**** *p* < 0.0001 vs. control; Figure 6A,B), which was abrogated in cells co-treated with L-proline (#### *p* < 0.0001 vs. AZE; Figure 6A,B). Similarly, pIRE1 reactivity was significantly increased upon exposure to AZE (** *p* < 0.01 vs. control; Figure 6A,C) and prevented by L-proline co-treatment (## *p* < 0.01 vs. AZE; Figure 6A,C). Noteworthy, L-proline treatment alone had no effects on either IBA1 or pIRE1 immunoreactivity (Figure 6A–C).

### 2.7. Effects of AZE Exposure and L-Proline Supplementation on Intracellular AZE Accumulation

To determine if the addition of AZE to the media increased the intracellular accumulation of the NPAA, and quantify its abundance in the absence or presence of L-proline, protein samples obtained from BV2 microglia that were treated or not treated with AZE alone or in combination with L-proline for 24 h were interrogated using triple/quadrupole mass spectrometry (TQMS).

As expected, Figure 7 shows that no AZE-bound proteins could be detected in control or L-proline-treated cells. Administration of AZE to the culture media (1000 μM) caused a robust increase in AZE-containing protein fractions (*** *p* < 0.01 vs. control; median = 254.4 ppm/mg protein; Figure 7). Co-supplementation of L-proline almost completely prevented AZE incorporation into proteins, with ~84% reduction in AZE-bound proteins (# *p* < 0.05 vs. AZE only, median abundance = 29.49 ppm/mg protein; Figure 7).

## 3. Discussion

In this study, we provide evidence that AZE treatment causes ER stress in both BV2 and primary microglia. We also demonstrate, for the first time, that co-administration of L-proline is sufficient to abrogate most of the detrimental effects of the toxic NPAA.

In recent work, we found that AZE induced both apoptotic cell death and the shift towards a pro-inflammatory phenotype in the BV2 microglial cell line [7]. Here, we provided additional evidence to define the mechanisms of action through which these biological responses occur, and identified a simple strategy to prevent the deleterious effects of AZE in resident immune cells of the CNS.

Microglial cells are the resident immune cells of the CNS, and play essential roles in the maintenance of homeostasis in the healthy [28] and diseased CNS [29,30,31]. Conditions that trigger chronic microglial polarization are critical for the development of oligodendropathies and neuronal damage, as seen in the CNS of MS patients [32,33,34], which cumulatively contribute to disease progression. Sobel’s study in AZE intoxicated mice identified signs of ER stress in the white matter of mice, suggesting that this pathogenic mechanism could be implicated in the damage occurring to myelin cells. Interestingly, we found that AZE treatment caused similar ER stress in microglial cells. ER stress in microglia (and perhaps in other immune cells) not only promotes cell damage, but is also able to trigger an inflammatory response [35,36] that can further contribute to a hostile CNS microenvironment, especially for the vulnerable neurons.

Due to AZE structurally similarity with L-proline, the NPAA evades editing by aaRS during protein assembly [9]. AZE takes advantage of such amino acid mimicry capability and, as demonstrated in this study, causes ER stress, likely due to protein misfolding of L-proline-rich proteins. AZE-induced ER stress has been previously described in NIH3T3 fibroblasts, COS-7 kidney and HEK-293 epithelial cells and HeLa cervical cancer cells [12,37]. However, knowledge on its effects in microglial cultures require further elucidation.

Based on the evidence from the literature, the UPR cascade that is activated in response to ER stress positively regulates the expression of pro-inflammatory markers in microglia and peripheral macrophages [38,39]. For instance, ATF4 stimulates the toll-like receptor 4 (TLR4)-mediated production of inflammatory cytokines such as IL-6 [40] and monocyte chemoattractant protein-1 (MCP-1) [41]. Other UPR-related molecules, such as C/EBP homologues protein (CHOP) and X-box binding protein 1 (XBP-1), participate in the induction of gene expression of various cytokines [38]. Furthermore, studies in BV2 microglia showed that pharmacological inhibition of the UPR activator PERK attenuated the HIV-1 transactivator protein-induced inflammatory M1 phenotype [42]. Additionally, ER stress induction using Brefeldin A—a potent inhibitor of vesicular transport between the ER and the Golgi apparatus—caused the activation of the NLR family pyrin domain containing 3 (NLRP3) inflammasome and increased the expression and release of interleukin 1β in microglial cultures [24], suggesting a direct crosstalk between the UPR and different inflammatory pathways in resident microglia.

These results prompted us to investigate if co-supplementation of the canonical amino acid L-proline would have been sufficient to prevent the toxic effects of AZE and halt the downstream ER stress cascade. Our decision to test the effects of co-administration, rather than pre-treating cells with L-proline prior to AZE exposure, was based on the rapid turnover of amino acids in cells to guarantee continuous renewal of the intracellular pool of these essential protein components [43]. Moreover, we sought to replicate in vitro an environmental model of toxicity, assuming that exogenous supplementation of L-proline (in the presence of AZE) would have prevented the ongoing toxicity caused by continued exposure to the toxic NPAA, mimicking the beneficial effects expected after the introduction of the correct amino acid through the diet in people environmentally exposed to the toxicant. Our data confirmed the almost complete reversal of AZE toxicity in BV2 microglial cells, including the de-activation of the UPR in both BV2 cells and primary microglia. At first glance, the latter findings may be counterintuitive, given that the UPR is needed to reverse the effects of ER stress [22,23]. However, mass-spectrometry experiments demonstrated that the co-administration of L-proline impedes most of the misincorporation of AZE into proteins of intoxicated cells, rather than blocking its toxic activity, suggesting that the constant presence of adequate intracellular levels of the canonical amino acid is sufficient to prevent overt ER stress and its pathogenic sequelae. This theory is supported by results, in which we show that L-proline had no effects on the expression of the anti-inflammatory marker CD206, suggesting that the amino acids prevent the phenotypic shift of microglia towards a pro-inflammatory phenotype, rather than reversing it. In fact, previous evidence demonstrated that mutant cells with overproduction of endogenous L-proline due to a mutation in the gene encoding for the enzyme responsible for the inter-conversion of L-glutamate to L-proline (namely pyrroline-5-carboxylate synthase, P5CS) are resistant to AZE toxicity [44], further supporting our theory.

L-proline exerts positive effects in the CNS [45,46]. However, in animal models, hyperprolinemia has also been associated with the occurrence of neurological deficits [15], likely due to energy metabolism deficits, Na(+),K(+)-ATPase, kinase creatine, oxidative stress and excitotoxicity [15]. Other studies indicate that excessive L-proline triggers oxidative damage in the blood cells of rats and to the liver [47,48], consistent with the idea that a balanced intake of the amino acid is required to provide optimal CNS protection. In this study, we co-administered L-proline at a concentration that was 20 times lower than AZE (50 µM L-proline vs. 1000 µM AZE). Still, we observed a minor activation of the ER stress machinery at 1 h post-treatment, which disappeared as early as after 6 h. Despite these findings, there are no apparent reasons to believe that L-proline itself elicited any significant detrimental effects on cultured microglia, as it is more likely that the initial (and mild) activation of UPR genes immediately after the L-proline co-administration may have been due to ER engulfment caused by the overabundance of the two amino acids, rather than an intrinsic noxious effect of L-proline. Alternatively, it is possible that, despite the obvious preference for L-proline over AZE during protein assembly, residual AZE may still be able to evade recognition by the ProRS at the earliest stages of treatment, enough to induce a minor ER stress and inflammatory response.

## 4. Materials and Methods

### 4.1. BV2 Microglial Cell Lines and Treatments

Murine BV2 microglial cells used in this study were grown in Dulbecco’s Modified Eagle’s Medium (DMEM)/F12 nutrient mixture (Sigma-Aldrich, Castle Hill, NSW, Australia) and were supplemented with 10% heat-inactivated fetal bovine serum (FBS, Scientifix, Clayton, VIC, Australia), 2 mM L-glutamine (Sigma-Aldrich, NSW, Australia), 100 U/mL penicillin and 200 μg/mL streptomycin (Sigma-Aldrich, Castle Hill, NSW, Australia). Cells were initially seeded in T25 flasks at a density of 1 × 10^6^ cells at 37 °C in a humidified atmosphere with 5% CO_2_ until 90–95% confluent (confirmed using light microscopy) prior to experimental treatments. Upon treatment, cells were passaged and plated either in 96-well plates (MTT and Griess assays), 12-well plates (IF) or 6-well plates (RNA and protein extraction) at the following densities: 5 × 10^4^ cells (96-well plates), 3 × 10^5^ cells (12-well plates) or 5 × 10^5^ cells (6-well plates). Media was replaced by either adding fresh DMEM/F12 alone (control), or media supplemented with 1000 µM AZE (cat n. AO760, Sigma-Aldrich, Castle Hill, NSW, Australia), 50 µM L-proline (cat. n. PO380, Sigma-Aldrich, NSW, Australia) or their combination for 1, 6 or 24 h, depending on the type of experiment. AZE concentration selected for this study (1000 µM) was based on dose-response experiments in BV2 and SH-SY5Y neuroblastoma cells [3,7]. L-proline concentration (50 µM) was chosen based on the findings by Song et al. in HeLa cells, where a 1:40 (L-proline/AZE) ratio provided partial protection to cells [9]. As such, we reduced this ratio to 1: 20 to test if full protection could be attained.

### 4.2. Primary Microglial Cells

Primary cultures of microglial cells were prepared as per the protocol described by Schildge et al. [49], with minor modifications.

Coating of tissue culture vessels/coverslips: Prior to cell dissociation, culture vessels or coverslips were incubated with poly-D-lysine (P6407, Sigma-Aldrich, Castle Hill, NSW, Australia) diluted in sterile milliQ H_2_O at a final dilution of 0.1 mg/mL for 20 min. The solution was then aspirated, and vessels were washed twice with sterile milliQ H_2_O and allowed to dry for at least 4 h at room temperature in a laminar flow hood.

Dissociation of mixed glial cultures: Whole brains of P0-P2 mouse pups were isolated and meninges were removed. Brains were then placed in sterile petri dishes and dissected in sterile-filtered Hanks’ balanced salt solution (cat. n. H9394, Sigma-Aldrich, Castle Hill, NSW, Australia), where olfactory bulbs and cerebella were discarded whilst cortices were collected for mixed glia cells isolation. Cortices were then cut into small pieces and transferred to 50 mL falcon tubes for trypsin/EDTA digestion (2.5% trypsin/EDTA for 30 min at 37 °C; Sigma-Aldrich, Castle Hill, NSW, Australia) and DNase I treatment (10 mg/mL, Sigma-Aldrich, Castle Hill, NSW, Australia), followed by centrifugation at 300 rpm × 5 min. Thereafter, triturated tissue was further dissociated by pipetting up and down (6–8 times) using a fire-polished sterile glass Pasteur pipette, until a single cell suspension was attained. Cells were then plated in a T75 flask at a density of 10–15 × 10^6^ cells for 10–11 days (or until 80–90% confluent), with media replacements on alternate days.

Isolation of primary microglial cells: After 10 days, two separate layers of cells are clearly distinguishable, with astrocytes firmly adhering on the bottom layer and microglia on the top layer. To isolate microglial cells, flasks containing mixed glial cultures were vigorously tapped and then put in an oscillator at room temperature (200 rpm × 6 h). Supernatants containing enriched microglia were then centrifuged at 1200 rpm × 5 min. Finally, the pellet was dispersed in full growth media and plated at the appropriate cell density for at least 3–4 days prior to downstream experiments.

### 4.3. MTT Assay

To assess cell viability, we used the cell proliferation kit I (MTT) (Sigma-Aldrich, Castle Hill, NSW, Australia) following the procedures described in previous work [50]. Cells were seeded into 96-well plates at a concentration of 1 × 10^4^ cells/well. DMEM containing 0.5 mg/mL 3-[4,5-dimethylthiazol-2-yl]-2,5-diphenyltetrazolium bromide (MTT) solution was added in each well. Following incubation for 6 and 24 h at 37 °C, the medium was removed, and 100 μL of solubilization solution was added. Formazan formed by the cleavage of the yellow tetrazolium salt MTT was measured spectrophotometrically by absorbance change at 550–600 nm using a microplate reader.

### 4.4. Griess Reagent Assay

To assess nitrous oxide levels, cells were seeded at 2 × 10^4^ cells per well in a 96-well plate and incubated at 37 °C with 5% CO_2_ until cells reached 80% confluence. Cells were treated for 6 and 24 h with control media, 1000 µM AZE or 50 µM L-proline. The supernatant was collected and placed into a new 96-well plate. A total of 100 µL of freshly prepared Griess reagent was then added to each well and incubated at room temperature for 15 min on a slow oscillation protected from light. Absorbance was measured at 540 nm using the TECAN Infinite M1000-PRO ELISA reader (Tecan Australia Pty Ltd., Port Melbourne, VIC, Australia). Optical density values from each group were recorded and reported as a percentage of control.

### 4.5. RNA Extraction and Protein Extraction

Total RNA was extracted using 1 mL TRI reagent (Sigma-Aldrich, Castle Hill, NSW, Australia) and 0.2 mL chloroform and precipitated with 0.5 mL 2-propanol (Sigma-Aldrich). Pellets were washed twice with 75% ethanol and air-dried. RNA concentrations and purity were calculated using NanoDrop™ 2000 (ThermoFisher Scientific, Waltham, MA, USA), where RNA was considered free from DNA contamination or phenols if A260/280 > 1.9 and A260/230 > 2.1, respectively [51]. Proteins were extracted using radioimmunoprecipitation assay (RIPA) buffer (Sigma-Aldrich, Castle Hill, NSW, Australia) containing 1 × protease inhibitor cocktail (cOmplete™, Mini, EDTA-free Protease Inhibitor Cocktail, Sigma-Aldrich, Castle Hill, NSW, Australia).

### 4.6. cDNA Synthesis and Real-Time qPCR Analyses

A total of 1 µg of total RNA was loaded in each cDNA synthesis reaction. cDNA synthesis was conducted using the T1000 thermal cycler (Bio-Rad, Gladesville, NSW, Australia) in a final volume of 20 µL. Each reaction contained 1 μg of RNA diluted in a volume of 11 µL, to which we added 9 µL of cDNA synthesis mix (Tetro cDNA synthesis kit) (Bioline, Gladesville, NSW, Australia). Samples were incubated at 45 °C for 40 min, followed by 85 °C for 5 min. Finally, cDNA samples were stored at −20 °C until use.

Real-time qPCR analyses were carried out as previously reported [52,53] with minor modifications. For each gene of interest, qPCRs were performed in a final volume of 10 μL, which comprised 3 μL cDNA, 0.4 μL milliQ water, 5 μL of iTaq Universal SYBR green master mix (BioRad, Gladesville, NSW, Australia) and 0.8 μL of the corresponding forward and reverse primers (5 μM, Sigma-Aldrich, Castle Hill, NSW, Australia) to obtain a final primer concentration of 400 nM. The primer sets used are shown in Table 1. Reaction mixtures were loaded in Hard-Shell^®^ 96-Well PCR Plates and loaded into a CFX96 Touch™ Real-Time PCR Detection System (Bio-Rad, Gladesville, NSW, Australia). Instrument settings were as follows: (1) 95 °C for 2 min, (2) 60 °C for 10 s, (3) 72 °C for 10 s, (4) plate read, and (5) repeat step 2 to 4 for 45 cycles. For the melting curve analyses, settings were (1) 65 °C for 35 s, (2) plate read, and (3) repeat step 1–2 for 60 cycles). To examine changes in expression, we analyzed the mean fold change values of each sample, calculated using the ΔCt method. PCR product specificity was evaluated by introducing a negative control in each experiment and by melting curve analysis, with each gene showing a single peak.

### 4.7. Immunofluorescence

BV2 microglial cells were seeded on poly-L-lysine coated (Sigma-Aldrich, Castle Hill, NSW, Australia) coverslips (22 mm Ø, Sarstedt, SA, Australia) at a density of 80,000 cells and allowed to adhere overnight. For primary microglial cells, cell density was 4.5 × 10^3^ cells/cm^2^. The next day, cells were fixed with pre-filtered 4% paraformaldehyde (PFA: 4% in PBS pH 7.4) (Sigma-Aldrich, Castle Hill, NSW, Australia). Coverslips were then washed three times with ice-cold PBS and permeabilized for 10 min in sterile PBS containing 0.25% Triton X-100 (Sigma-Aldrich, Castle Hill, NSW, Australia) for 10 min, followed by 3 × washes in PBS for 5 min. Non-specific binding of antibodies was attained by incubating coverslips with 1% BSA in PBST for 30 min. BV2 cells were then incubated in a humidified chamber overnight at 4 °C using a rabbit anti-PERK primary antibody (GeneTex, Irvine, CA, USA (GTX129275); diluted 1/250 in PBST and 1% BSA), whereas primary microglia was incubated with the mouse anti-IBA1 antibody (1/250; Sigma-Aldrich, Castle Hill, NSW, Australia) and the rabbit anti-phospho-IRE1α^(Ser724)^ (1/250; Sigma-Aldrich, Castle Hill, NSW, Australia). The next day, primary antibodies were removed with 3 × washes in PBS for 5 min. BV2 cells were then incubated with a goat anti-mouse AlexaFluor 594-conjugated secondary antibody (1/500; Abcam, Melbourne, VIC, Australia) in 1% BSA in PBST for 2 h at room temperature with gentle oscillation. Primary microglia were incubated with a mixture of two secondary antibodies: goat anti-mouse AlexaFluor 594-conjugated and goat anti-rabbit AlexaFluor 488-conjugated secondary antibodies (both diluted at 1/500; Abcam, Melbourne, VIC, Australia). Secondary antibodies were then removed by washing cells three times with PBS for 5 min. Counterstaining of nuclei was performed by applying VECTASHIELD^®^ Antifade Mounting Medium with DAPI (Abacus DX, Meadowbrook, QLD, Australia). Negative controls, in which the primary antibody was omitted, were included to ensure target specificity, and showed no signal (not shown). BV2 stained coverslips were then imaged using a Nikon Eclipse TS2 inverted microscope (magnification 20×), whereas primary microglia were imaged on Leica Stellaris 8 confocal fluorescence microscope (Leica Microsystems) equipped with a 63.5× oil-immersion objective.

### 4.8. Triple/Quadrupole Mass Spectrometry

Untreated, AZE-, L-proline or AZE + L-proline treated BV2 microglia (for 24 h) were harvested, lysed in 50 μL ice-cold RIPA buffer and extracted by probe sonication (Qsonica Q125 Sonicator, Adelab Scientific, Thebarton, South Australia) for 30 s at 50% power for three times, with 1 min interval on ice in between. Protein lysates were then cleared by centrifugation at 10,000× *g* × 12 min. Supernatant containing proteins were then quantified using the bicinchoninic acid assay (BCA; ThermoFisher Scientific) according to manufacturer’s protocol and measured using the TECAN infinite M1000-PRO ELISA reader at 562 nm. Thereafter, proteins were precipitated by adding trichloroacetic acid (TCA) to a final concentration of 10%. Samples were then left overnight at 4 °C and centrifuged at 10,000× *g* × 10 min at 4 °C to recover precipitated proteins. The protein pellet was then washed three times with cold 10% (*w*/*v*) TCA, and transferred to a glass vial for hydrolysis under vacuum overnight at 110 °C in 6 M hydrogen chloride (HCl). The hydrolyzed pellet was reconstituted with 200 µL of 20 mM HCl and filtered through a 0.2 μm filter. Samples were then kept at −80 °C until analysis. Chromatographic separation was performed using a Shimadzu Nexera X2 UHPLC, using a Shimadzu 8060 triple quadrupole mass spectrometer (TQMS) for detection. Separation was carried out on a Waters BEH Amide column (2.1 × 100 mm, 1.7 μm particle size), using a flow rate of 0.8 mL/min with a column oven temperature of 30 °C. Solvent A consisted of 80 mM ammonium formate in ultrapure water plus 0.6% formic acid (FA), solvent B consisted of acetonitrile (ACN) plus 0.6% FA. AZE was eluted using the following stepped gradient of solvent B: 0.00 min 90%, 3.50 min 90%, 5.50 min 80%, 9.25 min 80%, 9.30 min 70%, 11.20 min 70%, 11.20 min 90% and 14.00 min 90%. The injection volume was 5 μL. The TQMS was run with an ESI source in positive mode with the following source parameters: 0.1 kV interface voltage, 400 °C interface temperature, 225 °C desolation line (DL) temperature and 400 °C heat block, 3 L/min nebulizing gas flow, 17 L/min heating gas flow, and 3 L/min drying gas flow. Nitrogen was used for drying, heating and nebulizing gas, while argon was used for the collision gas. The MRM transitions for AZE ([M + H+ ACN] +) are 143.00 *m/z* → 102.10 *m/z* (CE: −8.0), 143.00 *m/z* → 56.10 *m/z* (CE: −23.2).

### 4.9. Statistical Analyses

Statistical analyses were performed using GraphPad Prism software v9.3 (GraphPad Software, La Jolla, CA, USA). After checking for normal distribution of data, time course analyses were computed using repeated measures ANOVAs followed by Tukey post hoc tests. Image data was analyzed by a one-way ANOVA and Tukey post hoc tests. TQMS analyses were analyzed using non-parametric testing (Mann–Whitney U test). *p*-values less than 0.05 were considered statistically significant.

## 5. Conclusions

In conclusion, this study provides critical mechanistic insights into the toxicity of AZE in cultured microglia. UPR activation in response to ER stress seems to be the major contributor to microglial polarization and death, and L-proline co-supplementation is sufficient to prevent such damaging effects. Altogether, this understudied environmental toxicant emerges as a novel potential risk factor for neuroinflammation. Although further in vivo preclinical and/or human studies are needed to substantiate our discoveries, our findings warrant additional monitoring of this toxic amino acid. Similarly, our in vitro data suggest that L-proline supplementation may prove to be an effective and simple-to-implement strategy to prevent AZE toxicity in the CNS.

## Figures and Tables

**Figure 1 molecules-28-04808-f001:**
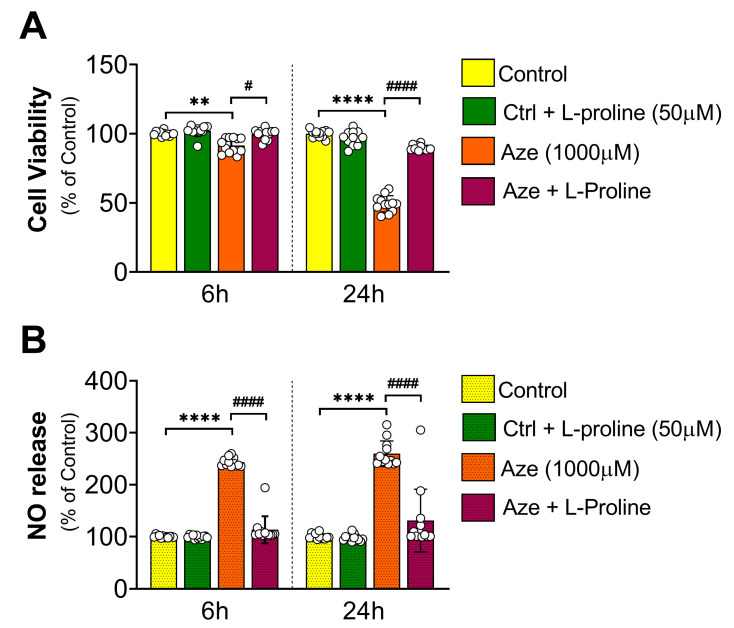
Effects of L-proline supplementation on cell viability and NO release in BV2 microglia exposed to toxic concentrations of AZE. (**A**) Cell viability and (**B**) NO release in untreated, L-proline-, AZE- or AZE + L-proline-treated cells at 6 and 24 h, as determined by MTT and Griess assays. Results are reported as % of control. AZE and L-proline concentrations are indicated in the corresponding legends. Data shown as the mean ± SEM from two independent experiments, each using six batches of cells (*n* = 6). ** *p* < 0.01 and **** *p* < 0.0001 vs. control or # *p* < 0.05 and #### *p* < 0.0001 vs. AZE-treated cells, as determined by repeated measures ANOVA followed by Tukey post hoc test. NO = nitric oxide; Ctrl = control; Aze = L-azetidine-2-carboxylic acid.

**Figure 2 molecules-28-04808-f002:**
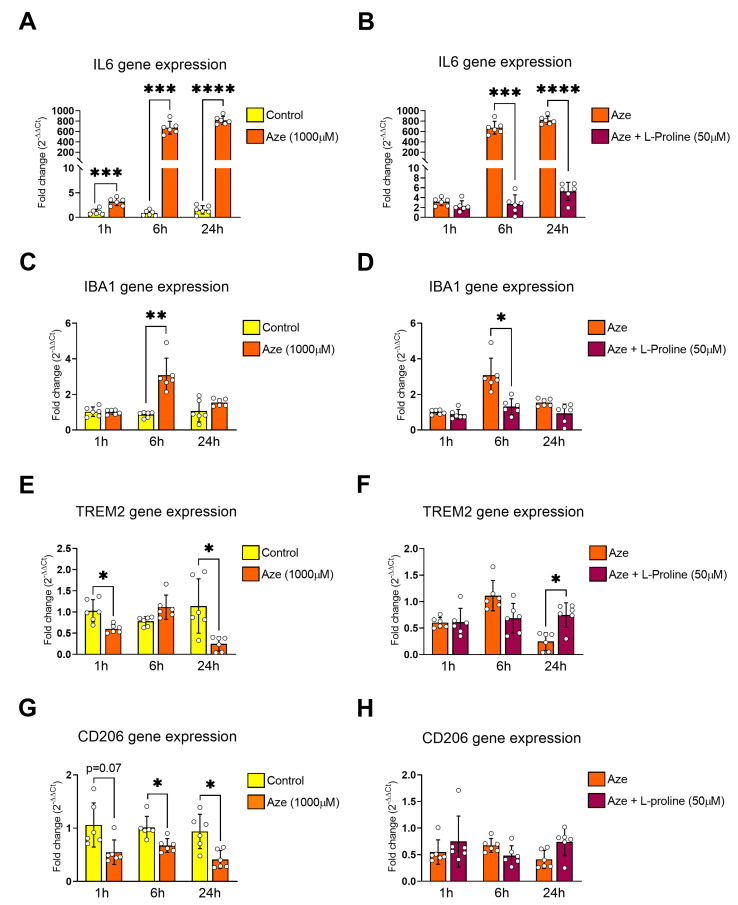
Effects of L-proline supplementation on the pro- and anti-inflammatory gene expression profile of BV2 cells after exposure to AZE. Gene expression in cells exposed to AZE alone or co-administered with L-proline was measured using real-time qPCR and quantified using the ΔCt method after normalization to the S18 housekeeping gene. Bar graphs display comparative gene expression changes in the following inflammation-related genes: (**A**,**B**) IL6, (**C**,**D**) IBA1, (**E**,**F**) TREM2 and (**G**,**H**) CD206. Results are presented as mean fold changes vs. control ± SEM (**A**,**C**,**E**,**G**) or vs. AZE (**B**,**D**,**F**,**H**) from *n* = 6 biological replicates. Statistically significant data (* *p* < 0.05, ** *p* < 0.01, *** *p* < 0.001 or **** *p* < 0.0001) were determined by repeated measures ANOVA followed by Tukey’s post hoc test. Aze = L-azetidine-2-carboxylic acid; IL6 = interleukin 6; IBA1 = ionized calcium binding adaptor molecule 1; TREM2 = triggering receptor expressed on myeloid cells 2; CD206 = cluster of differentiation 206 (also known as mannose receptor C-type 1).

**Figure 3 molecules-28-04808-f003:**
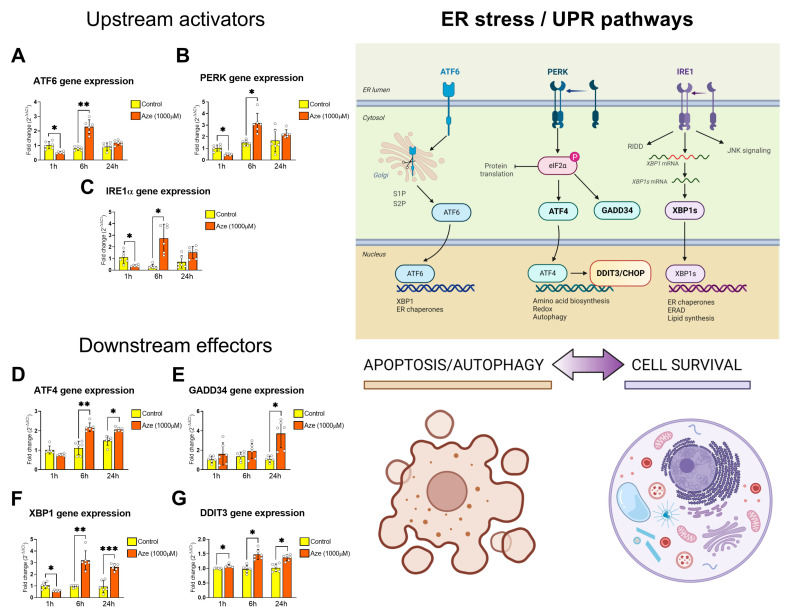
Effects of AZE treatment on the expression of ER stress/UPR genes. Gene expression was measured using real-time qPCR and quantified using the ΔCt method after normalization to the S18 housekeeping gene. Bar graphs show the differential expressions of the following upstream and downstream regulators of the ER stress/UPR response: ATF6, PERK (also known as EIF2AK3), IRE1α (also known as ERN1), ATF4, GADD34 (also known as PPP1R15A), XBP1 and DDIT3 (also known as CHOP). A schematic diagram showing the main ER stress/UPR pathways and biological outcomes (apoptosis/autophagy or survival) are shown on the right. Relative expression levels of (**A**) ATF6, (**B**) PERK, (**C**) IRE1α, (**D**) ATF4, (**E**) GADD34, (**F**) XBP1 and (**G**) DDIT3 transcripts were measured in untreated (control) and AZE-treated BV2 microglial cells at various times (1, 6 and 24 h, respectively). Results are presented as mean fold changes in controls ± SEM of three independent experiments, each run using two biological replicates per experiment (*n* = 6). * *p* < 0.05, ** *p* < 0.01 or *** *p* < 0.001 vs. control, as determined by repeated measures ANOVAs followed by Tukey post hoc tests. Aze = L-azetidine-2-carboxylic acid; ATF6 = activating transcription factor 6; PERK = protein kinase R-like endoplasmic reticulum kinase (also known as EIF2AK3 = eukaryotic translation initiation factor 2-alpha kinase 3); IRE1α = inositol-requiring transmembrane kinase/endoribonuclease 1α (also known as ERN1 = endoplasmic reticulum to nucleus signaling 1); ATF4 = activating transcription factor 4; GADD34 = growth arrest and DNA damage-inducible protein (also known as PPP1R15A = protein phosphatase 1 regulatory subunit 15A); XBP1 = X-box binding protein 1; DDIT3 = DNA damage inducible transcript 3 (also known as CHOP = C/EBP homologous protein); RIDD = regulated IRE1α-dependent decay; JNK = c-Jun N-terminal kinase.

**Figure 4 molecules-28-04808-f004:**
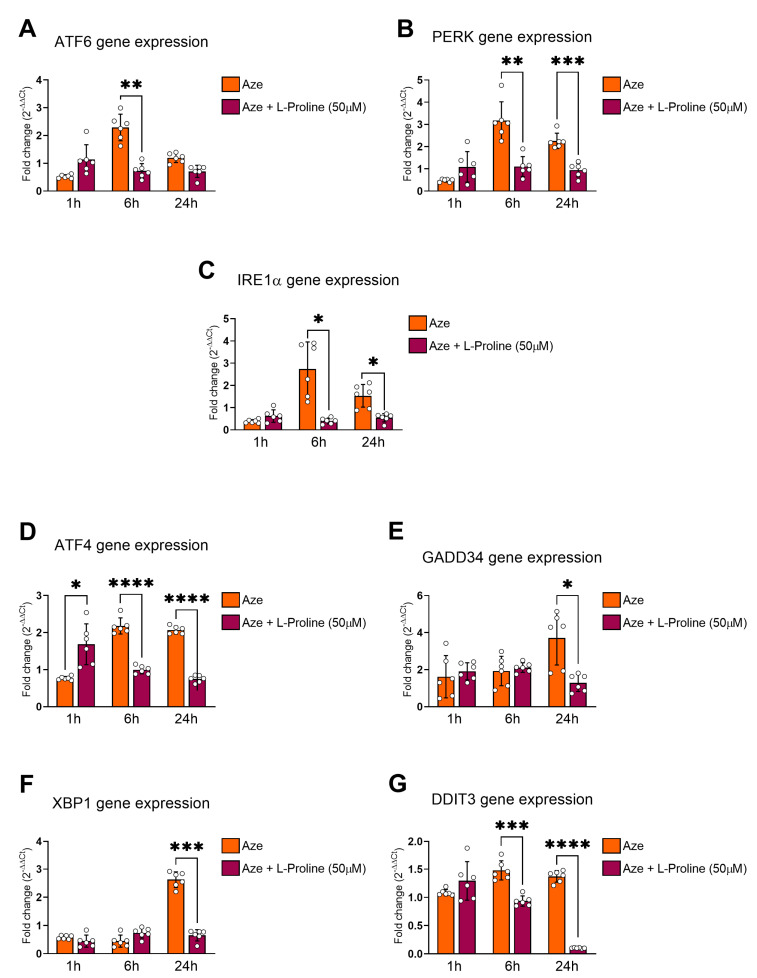
Effects of L-proline co-administration on the expression of ER stress/UPR genes in AZE-treated BV2 microglia. Gene expression was measured using real-time qPCR and quantified using the ΔCt method after normalization to the S18 housekeeping gene. Bar graphs show the differential expression of the following upstream and downstream regulators of the ER stress/UPR response: ATF6, PERK (also known as EIF2AK3), IRE1α (also known as ERN1), ATF4, GADD34, XBP1 and DDIT3. Relative expression levels of (**A**) ATF6, (**B**) PERK, (**C**) IRE1α, (**D**) ATF4, (**E**) GADD34, (**F**) XBP1 and (**G**) DDIT3 transcripts were measured in AZE-treated cells in the absence (Aze) or presence of L-proline (Aze + L-Proline) at various times (1, 6 and 24 h, respectively). Results are presented as mean fold changes in controls ± SEM of three independent experiments, each run using two biological replicates per experiment (*n* = 6). * *p* < 0.05, ** *p* < 0.01, *** *p* < 0.001 or **** *p* < 0.0001 vs. control, as determined by repeated measures ANOVA followed by Tukey post hoc test. Aze = L-azetidine-2-carboxylic acid; ATF6 = Activating transcription factor 6; PERK = protein kinase R-like endoplasmic reticulum kinase (also known as EIF2AK3 = eukaryotic translation initiation factor 2-alpha kinase 3); IRE1α = inositol-requiring transmembrane kinase/endoribonuclease 1α (also known as ERN1 = endoplasmic reticulum to nucleus signaling 1); ATF4 = activating transcription factor 4; GADD34 = growth arrest and DNA damage-inducible protein (also known as PPP1R15A = protein phosphatase 1 regulatory subunit 15A); XBP1 = X-box binding protein 1; DDIT3 = DNA damage inducible transcript 3 (also known as CHOP = C/EBP homologous protein).

**Figure 5 molecules-28-04808-f005:**
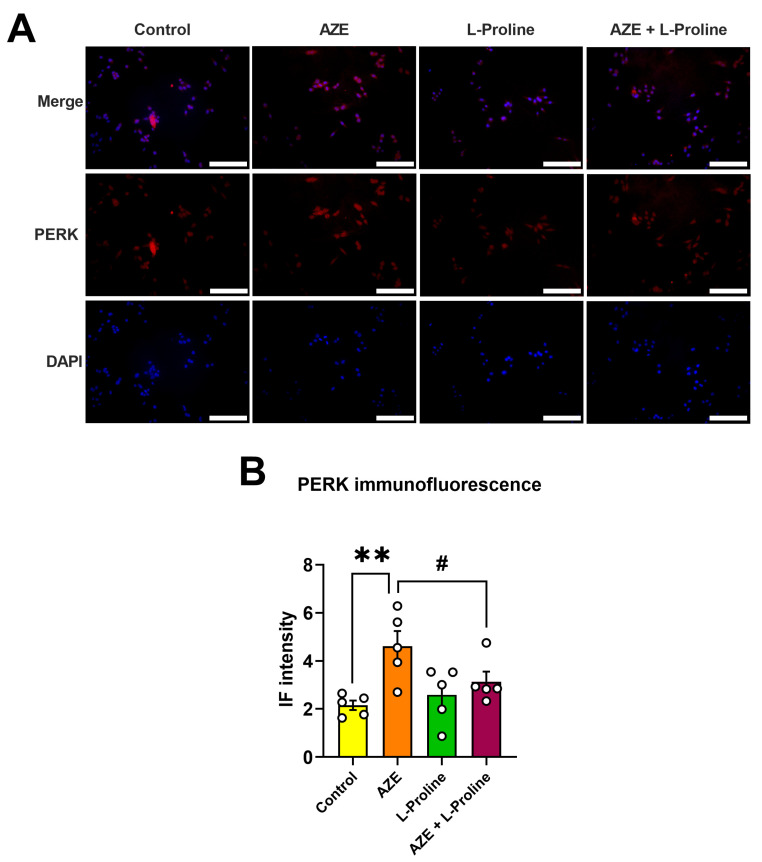
Effects of AZE treatment on PERK expression are prevented upon L-proline co-administration in BV2 microglia. (**A**) Representative photomicrographs showing PERK immunofluorescence (IF) and (**B**) related IF quantification of untreated cells (control) or following exposure to AZE (1000 µM), L-proline (50 µM) or their combination after 24 h. Images were taken on an Olympus BX51 Fluorescence Microscope using the 20× objective. Scale bar = 60 µm. (**B**) PERK immunofluorescence intensity was determined using NIH ImageJ version 1.52. At least five independent images (*n* = 5) were analyzed for each experimental condition. Results shown are the mean ± SEM. ** *p* < 0.01 vs. control or # *p* < 0.05 vs. AZE. Statistical significance was computed using a one-way ANOVA, followed by Tukey post hoc test. IF = immunofluorescence; Aze = L-azetidine-2-carboxylic acid; PERK = protein kinase R-like endoplasmic reticulum kinase.

**Figure 6 molecules-28-04808-f006:**
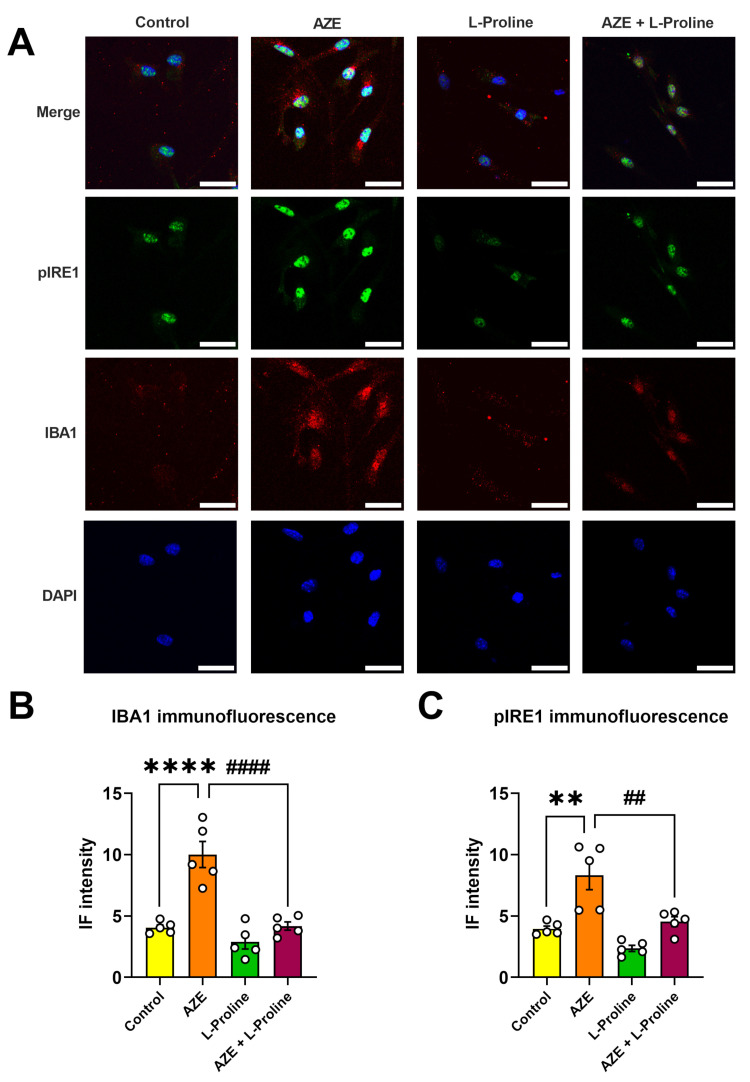
AZE-induced IBA1 and phospho-IRE1^Ser724^ (pIRE1) expression is prevented by L-proline co-administration in primary microglial cells. (**A**) Representative photomicrographs showing pIRE1 and IBA1 co-immunofluorescence (co-IF) and (**B**,**C**) related quantifications of untreated primary microglia (control) or following exposure to AZE (1000 µM), L-proline (50 µM) or their combination after 24 h. Images were taken on a Leica Stellaris 8 confocal fluorescence microscope (Leica Microsystems) using the 63.5× oil-immersed objective. Scale bar = 20 µm. (**B**) IBA1 and (**C**) pIRE1 IF intensities were determined using NIH ImageJ software, ver. 1.52. At least five independent images from independent experiments were quantified for each experimental condition (*n* = 5). Results shown are the mean ± SEM. ** *p* < 0.01 or **** *p* < 0.0001 vs. control; ## *p* < 0.01 or #### *p* < 0.0001 vs. AZE, as determined using a one-way ANOVA, followed by Tukey post hoc test. Aze = L-azetidine-2-carboxylic acid; IBA1 = ionized calcium binding adaptor molecule 1; pIRE1 = phospho-inositol-requiring transmembrane kinase/endoribonuclease 1α^Ser724^; DAPI = 4′,6-diamidino-2-phenylindole (nuclear dye).

**Figure 7 molecules-28-04808-f007:**
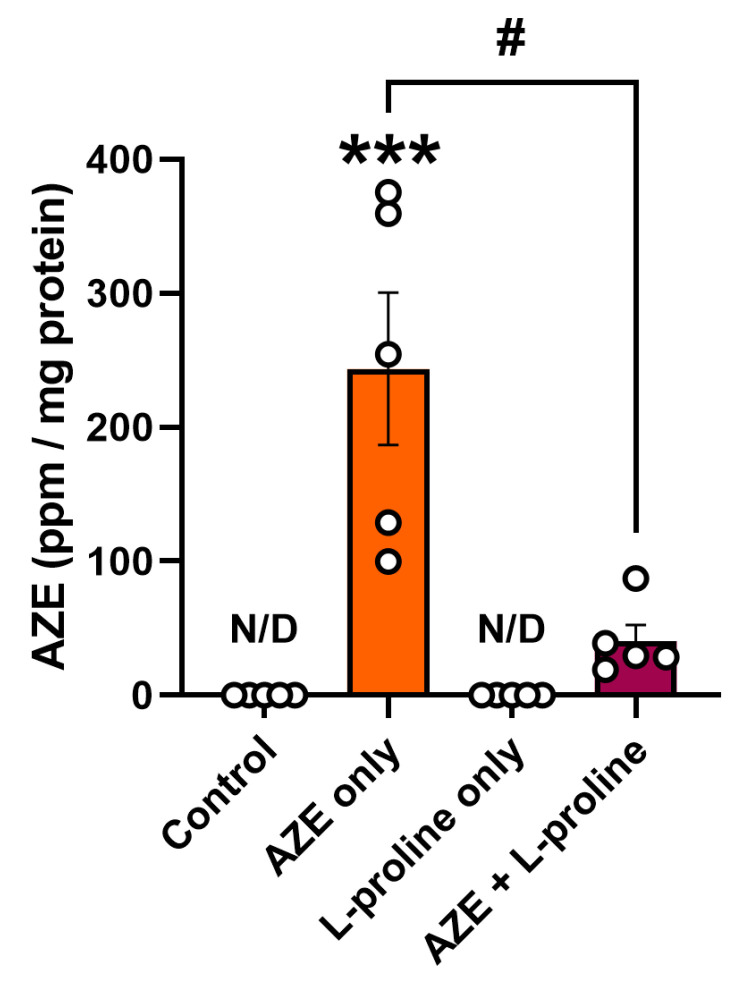
L-proline strongly reduces AZE misincorporation into proteins. BV2 microglial cells were either left untreated or treated with AZE only, L-proline only or their combination for 24 h. Thereafter, cells were lysed, and proteins were analyzed for AZE incorporation using triple/quadrupole mass spectrometry (TQMS). AZE abundance is reported as the mean ± SEM of five independent samples (*n* = 5). *** *p* < 0.01 vs. control or # *p* < 0.05 vs. AZE only. Statistical analyses were conducted using the Kruskal–Wallis test (non-parametric) followed by Dunn’s post hoc test, due to uneven variances across groups. N/D = not detected. Ppm = parts per million. AZE = L-azetidine-2-carboxylic acid.

**Table 1 molecules-28-04808-t001:** PCR primer sequences used to amplify the genes of interest. Forward and reverse primers, melting temperatures (Tm), predicted product size and accession numbers are shown.

Gene	Forward Sequence 5′-3′ Reverse Sequence 3′–5′	Tm (°C)	Product Size	Accession No.
ATF6	GAGCTGTCTGTGTGATGATAGT CTAGGTTTCACTCTTCGGGATT	59.88 59.90	94	NM_001081304.1
CD206	AGTGATGGTTCTCCCGTTTC ACCTTTCAGCTCACCACAAT	60.15 59.91	90	NM_008625.2
IRE1α (ERN1)	GAGACAAAGGAGAGTGTGTGAT TCAAGTAGTTCAGCTTGCTCTT	60.05 59.81	87	NM_023913.2
TREM2	CATCACTCTGAAGAACCTCCAA CTCCAGCATCTTGGTCATCTA	60.10 59.46	137	NM_031254.3
PERK (EIF2AK3)	CCTTGGTTTCATCTAGCCTCA ACTTGTAGGAAGATTCGAGCAG	59.95 60.12	156	NM_010121.3
XBP1	CAGAGAGTCAAACTAACGTGGT CAATGGCTGGATGAAAGCAG	60.21 59.89	160	NM_00127130.1
IBA1	GCTTTTGGACTGCTGAAGGC GTTTGGACGGCAGATCCTCA	60.04 61.45	114	NM_001361501.1
IL-6	CCCCAATTTCCAATGCTCTCC CGCACTAGGTTTGCCGAGTA	59.24 60.11	141	NM_031168.2
DDIT3	AGAAGGAAATGGAACGCACA CCAGCTGTGATGTGGGATAA	60.16 60.01	135	NM_008654.2
ATF4	CCTCAGACAGTGAACCCAAT AATGCTCTGGAGTGGAAGAC	59.89 59.91	127	NM_009716.3
GADD34	AGAAGGAAATGGAACGCACA CCAGCTGTGATGTGGGATAA	60.16 60.01	135	NM_008654.2
S18	CCCTGAGAAGTTCCAGCACA GGTGAGGTCGATGTCTGCTT	59.60 59.75	145	NM_011296.2

## Data Availability

All data generated or analyzed in this study has been included in this published article. Raw data can be made available upon reasonable request to authors.

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
