# Peer review of "L-Proline Prevents Endoplasmic Reticulum Stress in Microglial Cells Exposed to L-azetidine-2-carboxylic Acid"

_molecules, 2023, doi:10.3390/molecules28124808_

Round 1

Reviewer 1 Report

The authors investigated ifL-Azetidine-2-carboxylic acid (AZE) treatment caused endoplasmic reticulum stress in BV2 cells and tested the effects of L-proline co-administration in the presence of the imposter AZE. The authors found that treatment with AZE diminished cell viability and L-proline co-treatment completely reversed these effects. The nitric oxide (NO) levels were increased by AZE exposure and restored in the presence of L-proline. Activation of the unfolded protein response (UPR) genes (ATF4, ATF6, ERN1, PERK, XBP1, DDIT3, GADD34) in AZE-treated cells after 6 h was also found, along with changes consistent with M1 cell polarisation (increased IL-6, decreased CD206 and TREM2 expression). AZE-induced PERK induction was confirmed by immunocytochemistry. The triple/quadrupole mass spectrometry analyses demonstrated a robust increase in AZE-bound proteins after AZE treatment, which was reduced by 84% upon L-proline co-supplementation. The authors concluded that introduction of adequate levels of L-proline in the diet may prevent any harmful effects of environmental exposure to AZE in the CNS.

The manuscript is well organized and written, the experiments are adequately chosen and results are clear.

I have some suggestions for improvement the manuscript.

Title: Please, don’t use abbreviations in the title

Abstract: it should be more clear and concise. To much sentences begin with adverb.

Introduction: it should be insert meaning of abbreviation when it first appear in the text. For example line 57, CNS etc.

Discussion: The authors should begin with discussion of their own results, and than give a broader view in the context of the disease. Thus lines 318-333 are more appropriate for the larger context or for introduction section

Minor editing of English language is required.

Reviewer 2 Report

The authors based their study on the premise that L-azetidine-2-carboxylic acid (AZE) is a non-protein amino acid that shares structural similarities with the naturally occurring amino acid L-proline, allowing AZE to be erroneously incorporated into protein synthesis. Previously, it has been demonstrated that AZE induces both polarization and apoptosis in BV2 microglial cells. However, whether these effects involve endoplasmic reticulum (ER) stress remains unclear. Furthermore, whether L-proline co-administration prevents AZE-induced damage in microglia remains to be elucidated. In this study, they report the results of treating BV2 cells with AZE and its potential effect on ER proteostasis. In addition, they tested the effects of the co-administration of L-proline/AZE. BV2 cells were treated for 6 and 24 h with 50 µM L-proline, 1000 µM AZE, or both. Their results showed that AZE treatment decreased cell viability, and that co-administration of L-proline/AZE completely reversed this effect. In addition, they observed that the nitric oxide (NO) level increased with AZE but was restored in the presence of L-proline. Real-time qPCR analysis revealed the activation of unfolded protein response (UPR) genes in cells treated with AZE for 6 h, along with changes consistent with cell polarization. On the other hand, they observed AZE-induced PERK activation after 24 h of treatment, which was completely prevented by co-treatment with L-proline. Finally, mass spectrometry analyses showed an increase in AZE-bound proteins, which was reduced by co-treatment with L-proline. The authors concluded that through their study ER stress was identified as a pathogenic mechanism of AZE-induced microglial activation and death, which was reversed by the co-administration of L-proline. Furthermore, they suggested that including l-proline in the diet could prevent the deleterious effects of environmental exposure to AZE in regions where AZE-rich beets are included in the daily diet.

Although their findings impact the field, the following concerns weaken the study and should be addressed properly before resubmitting this paper.

1. The title is ambitious compared to the results generated in the study. Please correct it to a more realistic title.

2. Introduction. The background of this study is limited and has been poorly discussed. Therefore, a thorough review of the current state of this field is recommended, highlighting the hypothesis and uniqueness of the study as well as its contributions to the field.

3. The research methodology is standard and straightforward, with limited contribution to the technological development of the field. Please make a connection between the methods used and study hypothesis, considering the standards of scientific research.

4. The involvement of AZE (L-azetidine-2-carboxylic acid) as an inducer of ER stress has already been demonstrated. What is the uniqueness of this study? What are the novel contributions to the field? Please discuss extensively.

5. Please perform a proper discussion of previously generated knowledge in contrast to the findings reported in this study.

6. Finally, a thorough English language review is recommended, preferably by a proofreading service that employs native speakers with a scientific background/training.

Therefore, the current manuscript version is not endorsed for publication in Molecules.

A thorough English language review is recommended, preferably by a proofreading service that employs native speakers with a scientific background/training.

Reviewer 3 Report

The manuscript entitled “L-proline prevents ER stress in microglial cells exposed to L-azetidine-2-carboxylic acid – implications for multiple sclerosis pathophysiology” addresses the beneficial effect of L-proline against the toxicity in BV2 microglial cells and the associated molecular mechanisms. Initially, the authors proved that L-proline counteracted the AZE-triggered decline in BV2 viability and increased nitric oxide levels. Then, the authors proceeded to some implicated mechanisms. To this end, the authors demonstrated that L-proline co-supplementation attenuated the increased expression of unfolded protein response (UPR) genes, M1 polarization, and PERK protein expression. Together, these findings demonstrate that ER stress is a pathogenic mechanism for AZE-induced microglial toxicity, which is reversed by co-administration of L-proline. The current findings are interesting.

Comments:     

1) In the current study design, have the authors considered that the co-administration of L-proline with AZE may not be appropriate. In fact, the observed beneficial effects of L-proline may be driven by the competition between L-proline and AZE for the incorporation in some proteins culminating in protein misfolding.  The better design to confirm this point is the pre-treatment with AZE to microglial cells, followed by L-proline. Authors are advised to address this point and add the answers to the discussion section.

2) How were the in vitro concentrations selected including AZE and L-proline? The authors are advised to address this point and add the answers/proper citations to the material and methods section.

3) In real-time PCR analysis, have the authors considered that the gene expression assays of ATF4, ATF6, ERN1, PERK, XBP1, DDIT3, and GADD34 using RT-PCR may not be adequate for quantifying the target signals since the mRNA expression may not necessarily reflect the corresponding protein levels due to the post-translational modifications. In fact, detecting protein signals using ELISA or Western blotting is expected to give more reliable data than gene expression assays.

4) The qRT-PCR is missing biological (how many samples were used per experimental group) and technical repeat information (whether each sample was repeated during the assay). Moreover, did the authors check the RNA quality with A260/280, and perform an RT negative control to ensure no DNA contamination in the RNA extraction? Please, add these data in the relevant section in material and methods.

5) In the immunofluorescence and TUNEL assay, did the authors also perform a negative control to ensure the specific binding of the antibody to the target protein? Please, add the answer in the relevant section in Material and methods.

6) In the statistical analysis section, did the authors check data normality before proceeding to one-way ANOVA? Authors are advised to address this point and add the answers in the material and methods section.

7) In the statistical analysis of non-parametric data, since there are more than 2 treatments/groups (herein, 4), the authors are advised to analyze the data using the Kruskal-Wallis analysis of variance. When statistical significance is obtained, Dunn's test is applied. In fact, Mann Whitney test is performed for 2 treatments/groups only. The authors are advised to redo the statistical analysis for non-parametric data as described.

8) The conclusion stated in the abstract (lines 36-39) needs to be modified since additional animal and human studies are needed.

9) To make all figure legends stand-alone, authors are advised to add the full name of all the used abbreviations including AZE, IBA1, IRE, etc. at the end of each legend. Authors are advised to address this point and add the answers to the relevant figure legends.

10) The authors are advised to carefully revise the reference section. The authors are advised to unify the way they write the journal name. Sometimes it is written as a full name (such as references 1 and 13) while in other references it was written as an abbreviation. Please, follow the journal instructions in this regard. 

Minor editing of English language is required,

Round 2

Reviewer 2 Report

The improved version of this manuscript is endorsed for publication in "Molecules".

Minor editing is required.

Reviewer 3 Report

The authors adequately addressed the raised comments. Thanks!

Minor editing of the English language is required.